# Drug induced liver injury is associated with high mortality—A study from a tertiary care hospital in Pakistan

**Adeel Abid[1], Faryal Subhani[1], Farhana Kayani[2], Safia Awan[3], Shahab Abid[2]***

1 Medical College, Aga Khan University, Karachi, Pakistan, 2 Section of Gastroenterology and Department of Medicine, Aga Khan University, Karachi, Pakistan, 3 Department of Medicine, Aga Khan University, Karachi, Pakistan

* shahab.abid@aku.edu

## Abstract

### Background and aim

In light of few established drug induced liver injury (DILI) registries, this study aims to evaluate the clinical spectrum and predictors of mortality and morbidity of hospitalized patients with suspected DILI.

### Patients and methods

DILI cases were identified and categorized on basis of COIMS/RUCAM score and the exclusion of other liver diseases. Clinical and laboratory parameters were analyzed to identify the predictors of morbidity (prolonged hospital stay > 5 days) and mortality.

### Results

Out of 462 patients, there were 264 (57.6%) males and the mean age of the cohort was 50.83 years (range: 20–94 years). DILI was classified as definite or highly probable in 31.1%, probable in 62.5%, and possible in 7.4% of cases. Pattern of liver injury was hepatocellular in 25.1%, cholestatic in 56.17%, and mixed in 18.72% of patients. Anti-tuberculosis drugs (ATDs) were found to be the most common category of drugs causing DILI, in 295 (63.9%) patients. Clinically, encephalopathy was present in 21.6% patients; other presenting symptoms included abdominal pain (57.1%), vomiting (57.1%), jaundice (54.1%) and pruritus (42.3%). In-hospital mortality was 26.5% and prolonged hospital stay (> 5 days) was observed in 35.93% of patients. Mortality was significantly greater in patients with encephalopathy, male gender, hepatocellular pattern of DILI, increased INR and use of ventilator support.

### Conclusion

In our study, the most frequent cause of DILI in hospitalized patients was ATDs. More than a quarter of patients died during hospital stay. A close control of clinical and biochemical

**Data Availability Statement:** All relevant data are within the manuscript and its Supporting Information files.

**Funding:** This research project was fully sponsored by Ferozsons Private Limited, Drug Induced Liver

Injury Project (DILI-N); GC # CON000000000431. The funders had no role in study design, data collection and analysis, decision to publish, or preparation of the manuscript.

**Competing interests:** This research project was fully sponsored by Ferozsons Private Limited, Drug Induced Liver Injury Project (DILI-N); GC # CON000000000431. Moreover M/s Ferozsons Private Limited has no role related to employment, consultancy, patents, products in development, marketed products etc. related to this study. Furthermore to ensure that funding from Ferozsons does not alter our adherence to PLOS ONE policies on sharing data and materials.

parameters are required to prevent and monitor DILI, especially in patients taking ATDs in our region.

## Introduction

Drug induced liver injury (DILI) is defined as hepatotoxicity caused by various medications, herbs, or other xenobiotics, subsequently leading to abnormalities in liver tests or liver dysfunction with the reasonable exclusion of other etiologies [1]. Specific laboratory criteria are utilized to identify DILI: generally, a 3–5 times elevation of liver enzymes, namely transaminases (alanine aminotransferase [ALT] or aspartate aminotransferase [AST]), alkaline phosphatase (ALP), or bilirubin, above their upper limit of normal (ULN) is required [2]. Altogether, in excess of a thousand medicines and chemicals have been implicated in drug induced liver injury [3, 4].

In the United States, DILI accounts for nearly 10% of the total cases of acute hepatitis, 5% of all hospital admissions, and 50% of all cases of acute liver failures [5]. DILI carries a mortality rate of approximately 10% [3–5]. It is the premier reason for drug withdrawal by the Food and Drug Administration (FDA) in the United States [5, 6].

The wide spectrum of clinical symptomatology, non-availability of specific diagnostic markers and lack of standardization between studies performed to date make it difficult to establish causality to a particular drug. Causal association to a specific drug is not a straightforward matter, as it heavilydepends on exclusion of other causes (notably viral and autoimmune hepatitis) and temporal relationship of the drug to the derangement in patient's liver function tests (LFTs) [7]. As a result, sometimes certain scoring systems such as Roussel Uclaf Causality Assessment Method (RUCAM) [8], are used to assess the probability of association. The RUCAM system is a means of assigning points for clinical, biochemical, serologic and radiologic features of liver injury which gives an overall assessment score which reflects the likelihood that the hepatic injury is due to a specific medication [9].

Annual incidence of DILI ranges from 1.3 to 19 per 100,000 in various databases, depending on the country of origin, type of data and method of obtaining information [10–12]. The largest drug category responsible for DILI is antimicrobials, led by amoxicillin-clavulanate[13, 14]. Amongst antibiotics, ATDs are another major group associated with DILI especially in the developing world. Approximately 5.3% of all the cases in the United States DILI Network (US DILIN) were reported due to isoniazid (second only to amoxicillin-clavulanate) likewise 7% of the cases in the Spanish DILI Registry were due to isoniazid alone or in combination with other drugs [13, 14]. Other common drug groups include non-steroidal anti-inflammatory drugs (NSAIDs) [10, 14], herbal and dietary supplements (HDS) [15] and rarely statins [13, 16, 17].

A growing concern for pharmaceutical industry regarding drug development is hepatotoxicity induced by the newer molecular targeted agents (MTAs) which are increasingly being used in oncology. A third of patients treated with a protein kinase inhibitor experience liver injury, with pazopanib, sunitinib and regorafenib identified as the potentially lethal agents [18]. Similarly, 10% of patients treated with immune checkpoint inhibitors, such as ipilimumab, are susceptible to DILI [17]. Additionally, the epidermal growth factor receptor (EGFR) tyrosine kinase inhibitor (TKI) gefitinib is associated with 18.5% frequency of hepatotoxicity. It has resulted in casualties as well [19].

In many countries, DILI registries have been set up which record every DILI case with a formal causality determination process, providing in-depth information about the types of drugs that cause DILI, the pattern of injury and the risk of mortality and morbidity. There are

few established DILI registries in the region, and no centralized, national registry. This study aims to provide an analysis of clinical presentation and outcome of patients admitted with the discharge diagnosis of DILI from Pakistan.

## Patients and methods

### Ethics clearance

This study was reviewed and approved by ethics review committee of Aga Khan University (ERC-AKU).

### Study design

A retrospective cross-sectional study.

### Study setting and population

Patients admitted at Aga Khan University Hospital Karachi Pakistan, from January 2010 through December 2016, and discharged with a diagnosis of DILI, were recruited. The course of their hospital stay was reviewed through the medical record system.

### Inclusion criteria

Patients with suspected diagnosis of DILI with clear documentation of the possible drug implicated were included.

### Exclusion criteria

Patients with known or suspected acetaminophen toxicity, history of bone marrow or liver transplantation before the liver injury event, history of malignancy of liver and metastasis to liver, underlying hepatitis C virus (HCV), hepatitis B virus (HBV), or nonalcoholic fatty liver disease were all excluded alongside cases with other types of underlying chronic liver disease.

### Criteria for diagnosis of DILI

The diagnosis of DILI and the causal relationship between liver injury event and implicated drugs were evaluated in a formal and standardized fashion by using a causality instrument: Roussel Uclaf Causality Assessment Method (RUCAM) [9]. Points were awarded for seven components comprising of the following: time to onset of the injury following start of the drug, subsequent course of the injury after stopping the drug, specific risk factors (age, alcohol use, pregnancy), use of other medications with a potential for liver injury, exclusion of other causes of liver disease, known potential for hepatotoxicity of the implicated drug and response to re-challenge. The RUCAM provides a semi-quantitative evaluation of causality by assigning −3 to +3 points to each of the aforementioned seven components. Based on the final score, a causal relationship between the implicated agent and the liver injury event was categorized as highly probable (>8), probable (6–8), possible (3–5), unlikely (1 or 2), or excluded (<0).

### Criteria for assessment of clinical patterns of liver injury

According to the Council for International Organizations of Medical Sciences (CIOMS) criteria, DILI is classified as hepatocellular, cholestatic or mixed based on its *R*-value [9]. The *R*-value is defined as the serum ALT/ULN divided by the serum ALP/ULN ratio; *R*-values > 5 were classified as hepatocellular, < 2 as cholestatic and 2–5 as mixed injury [20].

## Criteria for severity assessment

The severity assessment was done according to the Chinese guidelines for the diagnosis and treatment of DILI in 2015 [21]. The severity was scored as follows:

1. Mild: serum enzyme elevations with total bilirubin (TBil) $< 2.5 \times$ ULN and International Normalization Ratio (INR) $< 1.5$.

2. Moderate: serum enzyme elevations and TBil $\geq 2.5 \times$ ULN or an INR $\geq 1.5$.

3. Severe: serum enzyme elevations and TBil $\geq 5 \times$ ULN with or without an INR $\geq 1.5$.

4. Acute liver failure: serum enzyme elevations and TBil $\geq 10 \times$ ULN or a daily elevation of TBil $\geq 17.1$ μmol/L, an INR $\geq 2.0$ and signs of hepatic or other organ failure related to DILI.

## Assessment of patient morbidity and mortality

In-hospital morbidity was quantified in terms of prolonged hospital stay, defined as hospital stay for more than 5 days. Predictors of mortality and morbidity were assessed by considering patients' clinical and laboratory parameters including liver synthetic functions (prothrombin time and serum albumin).

## Statistical analysis

The statistical analysis was conducted by using the Statistical Package for Social Science (SPSS) (Release 19.0 standard version, copyright © SPSS). A descriptive analysis was performed and results are presented as mean ± standard deviation for quantitative variables and numbers (percentages) for qualitative variables. To analyze the risk factors for poor outcome, the categorical variables were evaluated using the chi-square test while the means were compared by Student t-test. Factors predicting prolonged hospital stay were analyzed by multivariate logistic regression analysis. To establish statistical significance, p value $<0.05$ was considered significant.

## Compliance with ethical requirements

The study was undertaken upon receiving approval from Ethics Review Committee (ERC). Requirement for informed consent was waived by ERC. After completion of data collection by the authors, information was made anonymous for the statistician to proceed with data analysis.

## Results

A total of 462 DILI cases were identified (Fig 1), out of which 264 (57.6%) patients were male with a mean age of 50.83 (range: 20–94). By using the RUCAM model for drug causality assessment, DILI was classified as definite or highly probable in 141 (31.1%), probable in 289 (62.5%) and possible in 34 (7.4%) cases.

## Pattern of liver injury

Pattern of liver injury was hepatocellular in 116 (25.1%), cholestatic in 260 (56.17%) and mixed in 86 (8.72%) patients with a discharge diagnosis of DILI (Fig 2).

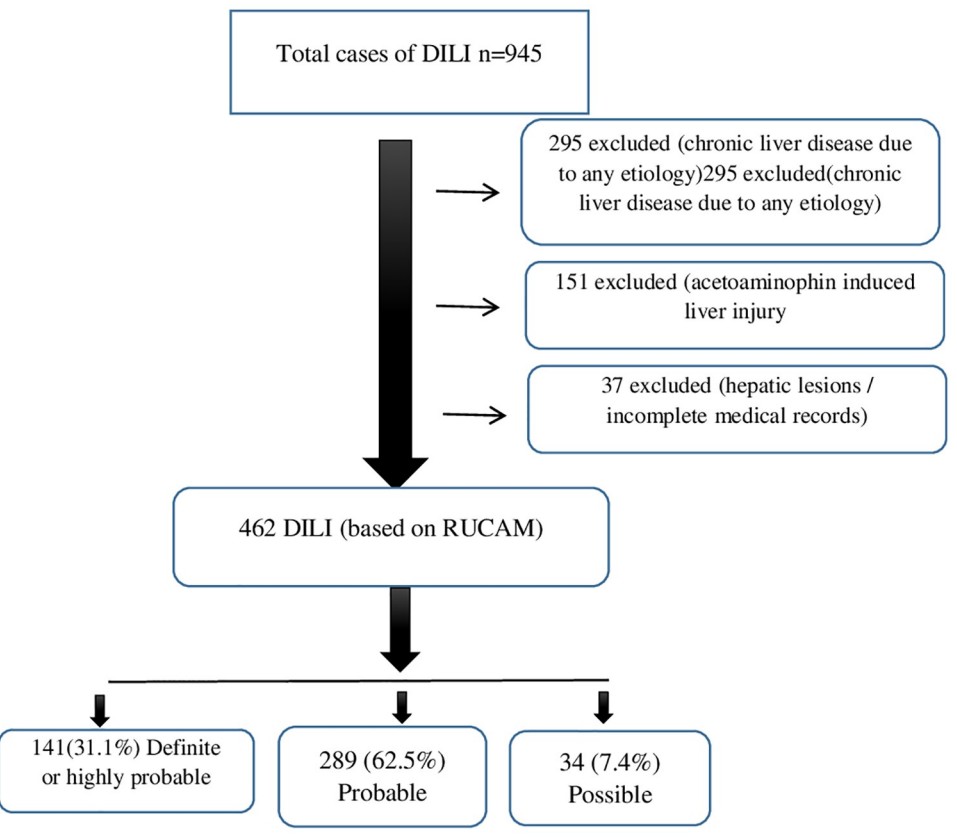

**Fig 1. Flow diagram showing selection DILI cases.**

## Severity of liver injury

The severity of liver injury was found to be mild in 204 (44%), moderate in 78 (16.8%), and severe in 54 (13.8%) patients, while 116 (25.1%) cases were seen to have had liver failure due to drug intake. Mortality was significantly high in patients with liver failure (p value = 0.006). Table 1.

## Presenting features of patients with DILI

Encephalopathy was present in 98 (21.6%) patients on the day of hospital admission while patients who presented with abdominal pain, vomiting, jaundice and pruritus were in the following order: 57.1%, 57.1%, 54.1%, and 42.3% (Fig 3).

Furthermore, mean total bilirubin levels, ALT and AP levels were 5.37mg/dl (range: 0.20–79.1), 358.65(range: 7–8938) IU/L and 168.68(range: 32–1040) IU/L respectively.

## Drug categories causing DILI

The top three causes of DILI in our study were anti-tuberculosis drugs (ATDs) followed by homeopathic or herbal medications and antiarrhythmic drugs. Patients taking ATDs in combination with different medications are listed (Table 2).

Other drugs implicated are displayed on Fig 4.

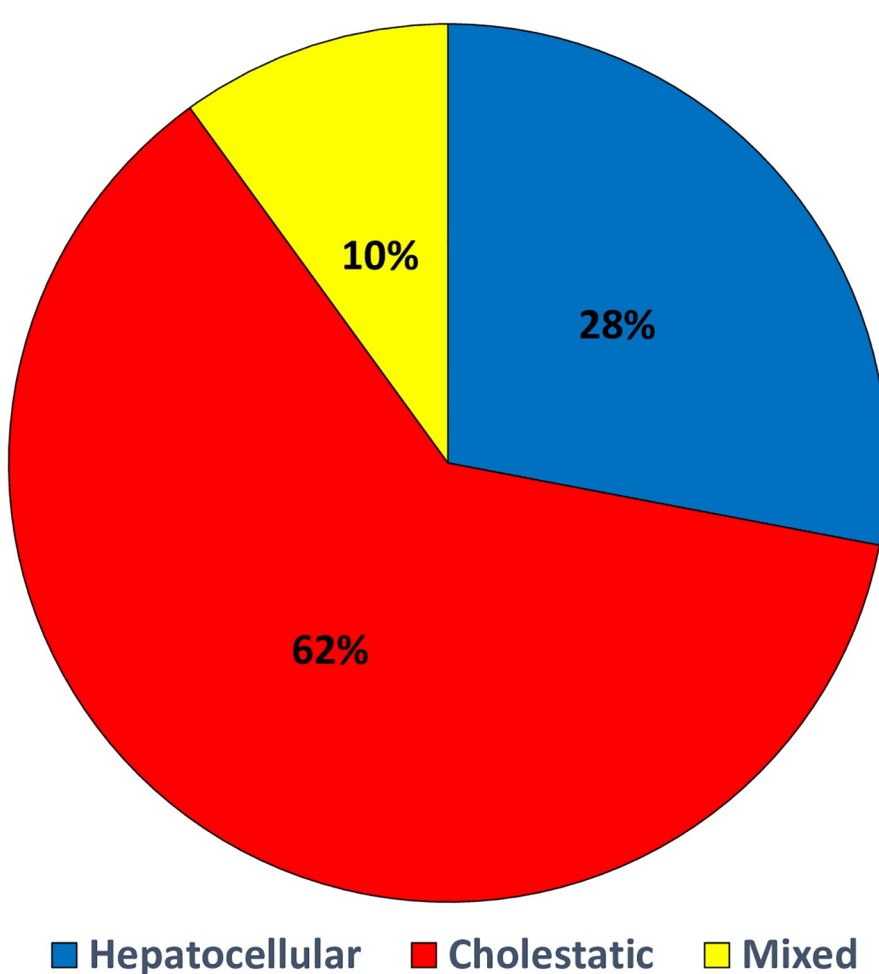

**Fig 2. Type of drug induce liver injury (n = 462).**

**Table 1. Severity of DILI and relationship of age and gender with mortality (n = 462).**

|  | Mild n = 204(44.2%) | Moderate n = 78(16.9%) | Severe n = 64(13.9%) | ACLF n = 116(25.1%) | p value |
|---|---|---|---|---|---|
| Mortality |  |  |  |  |  |
| Yes | 42(20.6) | 20(25.6) | 16(25) | 44(38.6) | 0.006 |
| No | 162(79.4) | 58(74.4) | 48(75) | 70(61.4) |  |
| Age |  |  |  |  |  |
| ≤35 years | 46(22.5) | 16(20.5) | 12(18.8) | 28(24.1) | 0.19 |
| 36–45 | 36(17.6) | 14(17.9) | 8(12.5) | 20(17.2) |  |
| 46–55 | 40(19.6) | 10(12.8) | 12(18.8) | 26(22.4) |  |
| 55–65 | 40(19.6) | 18(23.1) | 12(18.8) | 30(25.9) |  |
| >65 yrs | 42(20.6) | 20(25.6) | 20(31.3) | 12(10.3) |  |
| Gender |  |  |  |  |  |
| Male | 102(50) | 52(66.7) | 50(78.1) | 62(53.4) | <0.001 |
| Female | 102(50) | 26(33.5) | 14(21.9) | 54(46.6) |  |

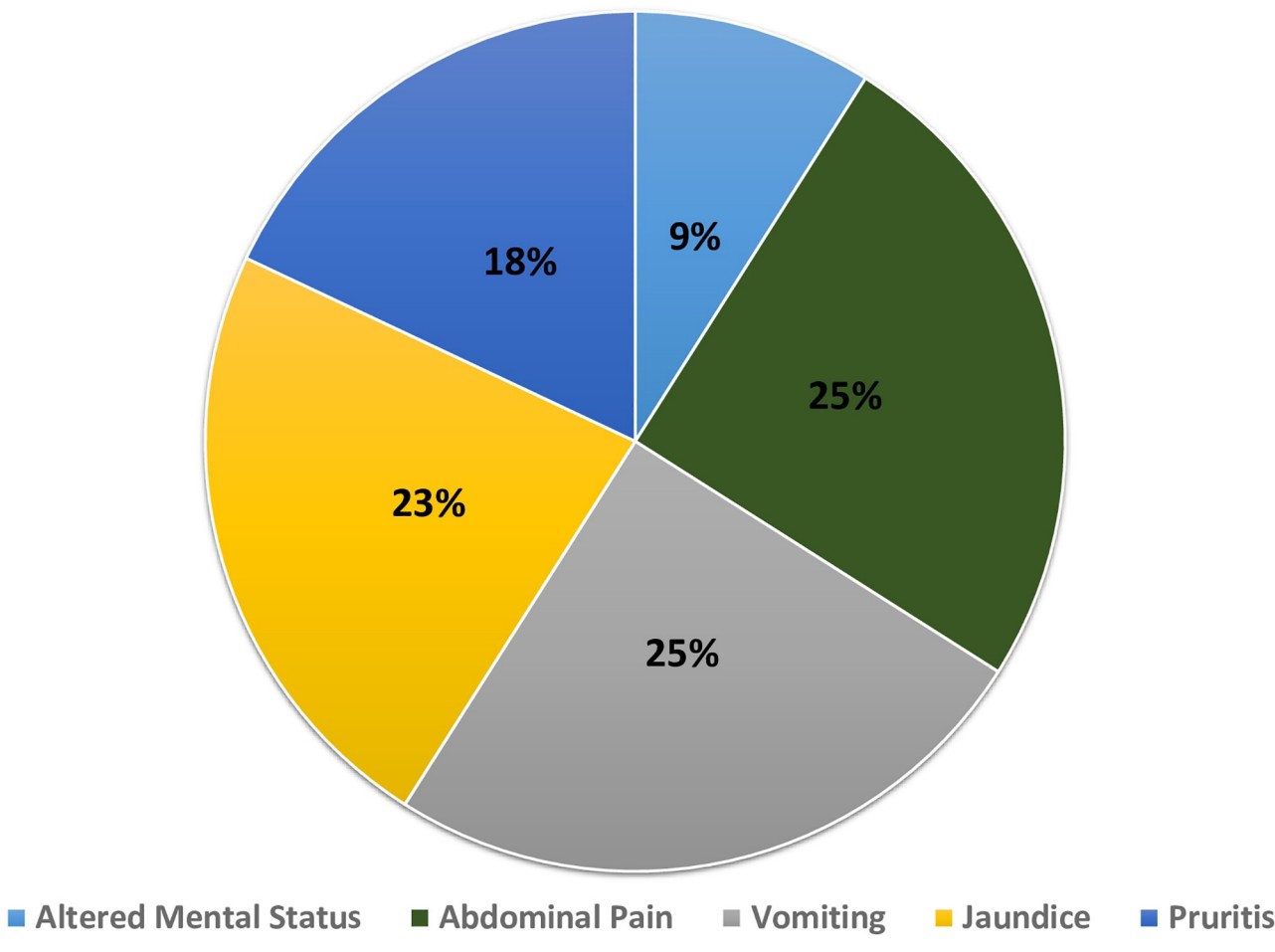

**Fig 3. Presenting features of patients with drug induced liver injury (n = 462).**

## Predictors of mortality and morbidity

All patients were managed as per standards of care, essentially supportive treatment was administered. A subset of patients with severe DILI received N- acetyl cysteine, and a subset of patients required ventilator support. In-hospital mortality was 122 out of 462 (26.5%) and morbidity (quantified as prolonged hospital stay more than 5 days) was observed in 214 out of 462 (35.93%) patients. None of the patients underwent a liver transplant due to non-availability of the facility at our institution and in the city during that time period.

**Table 2. Patients on anti-tuberculosis drugs.**

| Combination of drugs | Number of patients n = 295 (%) |
|---|---|
| ATDs alone | 182(61.6) |
| ATDs with NSAID | 26(8.8) |
| ATDs with antibiotic | 83(28.1) |
| ATDs with antiepileptics | 4(1.3) |

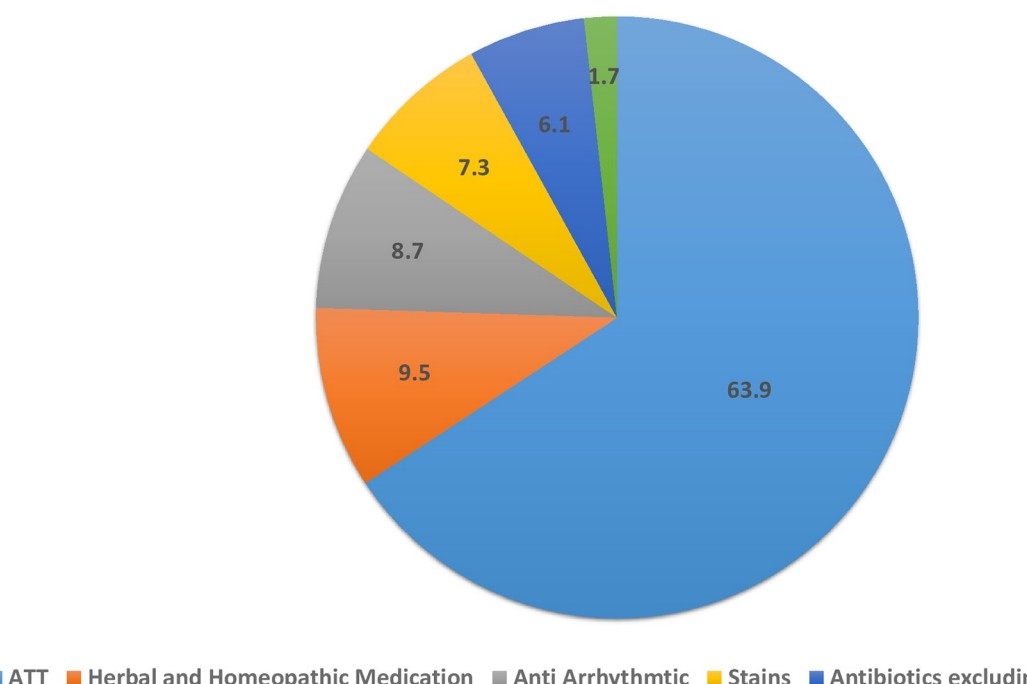

**Fig 4. Drug categories causing drug induced liver injury (n = 462).**

On multivariate analysis, mortality was significantly greater in patients with encephalopathy, male gender, hepatocellular pattern of DILI, increased INR ($>1.5$), acute liver failure and patients who were on ventilator support in ICU. Table 3.

Likewise, prolonged hospital stay (duration of $>5$ days) was associated with female gender, increased ALT, AST aspartate aminotransferase levels, use of ventilator support and mixed pattern of DILI. Table 4.

## Discussion

Drug induced liver injury is the most under-recognized and under-reported cause of liver injury, ultimately leading to underestimation of its burden. The present study analyzes hospitalized patients suffering from drug induced liver injury who were admitted in a tertiary care center in Pakistan, over a seven-year period. This is a large data set related to DILI from a developing country from where there is paucity of such kind of information.

One important finding from this study is that the order and frequency of drugs associated with DILI is different from the list provided in the report from the Drug Induced Liver Injury Network (DILIN) and the Spanish Registry [13, 14]. These studies showed that amoxicillin-clavulanate was the most common causative agent amongst the antimicrobials. A recently published review found 9 of the top 10 causes of DILI to be antibiotics; this is a measure of their hepatotoxic potential, as well as the common use and duration of treatment with these drugs [20, 22].

We found ATDs to be the most commonly implicated drug with approximately 64% of cases that were reviewed having received ATDs. This likely reflects the differences in the epidemiology of infectious diseases and corresponded to numbers observed by other studies from this region [23]. ATDs were followed by homeopathic and herbal medications with 9% of cases having received it, similar to other prior studies [24]. After ATDs, the category of drugs most frequently

**Table 3. Predictors of mortality of drug induced liver injury (n = 462).**

| Patient characteristics | Death (n = 122) | Survival (n = 340) | *p* value |
|---|---|---|---|
| Age, in years | 53.3 ± 15.1 | 49.8 ± 17.0 | 0.03 |
| Gender | | | |
| Male | 80(65.6) | 184(54.4) | 0.03 |
| Female | 42(34.4) | 154(45.6) | |
| DM | 30(24.6) | 70(20.7) | 0.22 |
| Dyslipidemia | 50(41) | 108(32) | 0.07 |
| ATT | 70(57.4) | 224(66.7) | 0.08 |
| Antibiotic | 8(6.6) | 20(5.9) | 0.80 |
| Antiepileptic | 0 | 8(2.4) | 0.08 |
| Antifungal | 10(8.2) | 16(4.7) | 0.15 |
| Atorvastatin | 10(8.2) | 22(6.5) | 0.57 |
| Chemotherapy | 10(8.2) | 4(1.2) | <0.001 |
| Herbal | 10(8.2) | 32(9.5) | 0.63 |
| Antimalarials | 6(5.6) | 4(1.4) | 0.02 |
| Digoxin | 0 | 12(4.3) | 0.006 |
| Antidepressants | 2(2.0) | 14(5.0) | 0.17 |
| Altered Mental Status | 46(37.7) | 52(15.4) | <0.001 |
| Jaundice | 74(60.7) | 174(51.5) | 0.08 |
| Pruritus | 78(63.9) | 184(54.4) | 0.06 |
| Abdominal pain | 120(98.4) | 326(96.4) | 0.26 |
| N acetyl cysteine | 38(31.1) | 54(16) | <0.001 |
| Hospital stay in days | 10.7 ± 10.9 | 6.7 ± 6.5 | <0.001 |
| Intubation | 32(38.1) | 18(7.0) | <0.001 |
| TB | 6.9 ± 12.8 | 4.8 ± 7.0 | 0.09 |
| IB | 2.5 ± 5.7 | 1.08 ± 1.4 | 0.007 |
| PT | 19.4 ± 13.0 | 15.0 ± 7.0 | <0.001 |
| INR | 1.93 ± 1.3 | 1.45 ± 0.65 | <0.001 |
| AP | 202.7 ± 183.1 | 165.3 ± 121.2 | 0.03 |

implicated in DILI was homeopathic and herbal medications, with a frequency within a range provided in prior studies from regions with a history of common consumption [24].

More than 20% of patients in our series had encephalopathy accounting for fulminant or acute liver failure at the time of presentation in the hospital. Conversely, the Spanish registry reported very low number of patients with fulminant hepatic failure with 11 out of 439 cases being classified as such [14, 25]. The high prevalence of encephalopathy in our study can be attributed to a delay in presentation to the hospital with very little knowledge about the drug being a cause of liver injury. Additionally, our center is one of the main tertiary care hospitals in Pakistan that receives an increasing number of complicated referrals: this may have resulted in more serious clinical presentations and contributed to greater DILI cases arising due to ATDs.

Another noteworthy observation deduced from our study is the fact that more than a quarter of hospitalized patients with DILI died while in the hospital. The mortality rate in our study appeared significantly high compared to that observed in several other studies, which ranges from 10 to 17.3% [10, 13, 26, and 27]. This difference in mortality is perhaps due to the fact that our series of DILI is for hospitalized patients which are expected to be more severely ill. Another factor for high mortality in our study could be the fact that ATDs was the leading cause of DILI as it has been observed in an Indian study that mortality in DILI patients on ATDs was significantly high compared to those not taking ATDs: 21.5% vs. 11.4% respectively

**Table 4. Predictors of prolonged hospital stay (>5 days) of patients with DILI (n = 462).**

| Patient characteristics | < 5 days (n = 248) | >5 days (n = 214) | *p* value |
|---|---|---|---|
| Age, in years | 50.2 ± 16.8 | 51.3 ± 16.4 | 0.45 |
| Gender | | | |
| Male | 154(62.1) | 112(52.6) | 0.03 |
| Female | 94(37.9) | 101(47.4) | |
| DM | 48(19.4) | 31(14.6) | 0.17 |
| Dyslipidemia | 51(20.6) | 50(23.5) | 0.45 |
| ATT | 43(76.8) | 37(58.7) | 0.03 |
| Antibiotics | 4(7.0) | 4(6.3) | 0.88 |
| Antiepileptics | 2(3.5) | 1(1.6) | 0.50 |
| Antifungal | 1(1.8) | 3(4.8) | 0.35 |
| Amiodarone | 5(8.8) | 5(7.9) | 0.86 |
| Statins | 6(10.5) | 5(7.9) | 0.62 |
| Chemotherapy | 1(1.8) | 3(4.8) | 0.38 |
| Herbal | 4(7.0) | 7(11.1) | 0.45 |
| Antimalarials | 0 | 2(4.8) | 0.49 |
| Mortality | 27(22.1) | 34(32.7) | 0.07 |
| N acetyl cysteine | 21(17.4) | 26(25) | 0.16 |
| History of alcohol | 1(1.8) | 2(3.3) | 0.58 |
| Intubation | 7(12.5) | 15(25) | 0.08 |
| TB | 4.9 ± 9.2 | 6.0 ± 11.4 | 0.25 |
| DB | 2.6 ± 2.3 | 5.1 ± 8.6 | 0.03 |
| IB | 1.11 ± 1.2 | 2.72 ± 5.9 | 0.04 |
| GGT | 151.3 ± 138.0 | 129.8 ± 149.9 | 0.11 |
| SGPT | 308.6 ± 630.9 | 439.1 ± 1093.0 | 0.12 |
| AP | 177.0 ± 125.8 | 165.7 ± 135.7 | 0.35 |
| SGOT | 400.1 ± 765.9 | 695.8 ± 1416.5 | 0.17 |
| R ratio | 9.6 ± 23.8 | 11.6 ± 21.7 | 0.34 |

(p = 0.02)[28]. Lack of facilities for liver transplantation could be another reason for high mortality in our series.

Very few studies have reported predictors of outcome for DILI which include hepatocellular damage, high bilirubin and female sex, as described by the US DILI network [15]. The Spanish registry and a Swedish study have described the hepatocellular pattern of damage as the most common form of liver injury associated with high incidence of liver transplantation or death if patient with jaundice [14, 27].

In a Chinese study, ATDs were found to be the primary etiological factor for fatal DILI. Additionally, the same study also identified that hepatic encephalopathy, ascites, jaundice, alcohol abuse and direct bilirubin levels were associated with the death of DILI patients [29]. Likewise in an Indian study, high-MELD score or a combination of ascites, encephalopathy, high bilirubin, prothrombin time, and leukocyte count were identified as predictors of mortality [28]. In our study, we also observed that mortality was significantly greater in patients with encephalopathy, male gender, hepatocellular pattern of DILI, increased INR (>1.5) and patients on ventilator support.

Limitations of the present study include a retrospective study design and a sample population based in a single tertiary care center setting. Non-availability of transplantation facility for ultimate treatment of patients restricted us from reviewing the outcomes in such patients in detail.

## Conclusion

In the present study, ATDs was seen to be the most frequent cause of DILI in hospitalized patients. More than a quarter of patients died during hospital stay. As a result, care among physicians is required while prescribing potentially hepatotoxic agents. A close control of clinical and biochemical parameters is required while prescribing potentially hepatotoxic agents, especially ATDs in our region. Additionally, efforts at the national level should be undertaken to create greater public awareness about DILI especially while using ATDs.

## Supporting information

**S1 Data.**
(RTF)

## Author Contributions

**Conceptualization:** Shahab Abid.

**Data curation:** Adeel Abid, Faryal Subhani, Farhana Kayani.

**Formal analysis:** Safia Awan.

**Investigation:** Adeel Abid, Faryal Subhani, Farhana Kayani.

**Methodology:** Farhana Kayani, Safia Awan, Shahab Abid.

**Project administration:** Shahab Abid.

**Supervision:** Farhana Kayani.

**Writing – original draft:** Adeel Abid, Faryal Subhani, Farhana Kayani.

**Writing – review & editing:** Shahab Abid.

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
