## [Decision Letter · Decision Letter 0]

10 Feb 2020

PONE-D-19-34738

Drug induced liver Injury is associated with high mortality - a study from a tertiary care hospital in Pakistan

PLOS ONE

Dear Dr. Abid,

Thank you for submitting your manuscript to PLOS ONE. After careful consideration, we feel that it has merit but does not fully meet PLOS ONE’s publication criteria as it currently stands. Therefore, we invite you to submit a revised version of the manuscript that addresses the points raised during the review process.

We would appreciate receiving your revised manuscript by Mar 26 2020 11:59PM. To enhance the reproducibility of your results, we recommend that if applicable you deposit your laboratory protocols in protocols.io, where a protocol can be assigned its own identifier (DOI) such that it can be cited independently in the future. For instructions see: http://journals.plos.org/plosone/s/submission-guidelines#loc-laboratory-protocols

We look forward to receiving your revised manuscript.

Kind regards,

Chiara Lazzeri

Academic Editor

PLOS ONE

Journal Requirements:

2. In the ethics statement in the manuscript and in the online submission form, please provide additional information about the patient records used in your retrospective study. Specifically, please ensure that you have discussed whether all data were fully anonymized before you accessed them and/or whether the IRB or ethics committee waived the requirement for informed consent. If patients provided informed written consent to have data from their medical records used in research, please include this information.

3. We noticed minor instances of text overlap with the following previous publications, which need to be addressed:

https://doi.org/10.1016/j.clinre.2017.08.003

https://doi.org/10.1016/j.cld.2016.08.004

In your revision please ensure you cite all your sources (including your own works), and quote or rephrase any duplicated text outside the methods section. Further consideration is dependent on these concerns being addressed.

5. Please upload a new copy of Figure 2-4 as the details are not clear. Please follow the link for more information: http://blogs.PLOS.org/everyone/2011/05/10/how-to-check-your-manuscript-image-quality-in-editorial-manager/

6. Thank you for stating the following financial disclosure:

"Ferozsons Private Limited, Drug Induced Liver Injury Project (DILI-N) having GC # CON000000000431"

Reviewers' comments:

Reviewer's Responses to Questions

**Comments to the Author**

1. Is the manuscript technically sound, and do the data support the conclusions?

Reviewer #1: Partly

Reviewer #2: Partly

2. Has the statistical analysis been performed appropriately and rigorously? 

Reviewer #1: N/A

Reviewer #2: Yes

3. Have the authors made all data underlying the findings in their manuscript fully available?

Reviewer #1: Yes

Reviewer #2: No

4. Is the manuscript presented in an intelligible fashion and written in standard English?

Reviewer #1: Yes

Reviewer #2: Yes

5. Review Comments to the Author

Reviewer #1: Authors evaluated the clinical spectrum and predictors of mortality and morbidity of hospitalized patients with suspected DILI in a medical facility in Pakistan. The most frequent cause of DILI is ATDs in hospitalized patients in their study, which is different from other registries. Authors suggested that this was due to differences in the epidemiology of infectious diseases in their region. The manuscript is well written, but authors should provide more information as suggested below.

1. Why did authors exclude patients with acetaminophen toxicity when acetaminophen is a known major cause of DILI?

2. Authors state that their results are different from other drug registries that have examined the most frequent-causing DILI drugs due to differences in infectious disease epidemiology. Authors did not show any sort of correlation for DILI-causing drugs and diseases present in previously studied regions compared to their study/region. How can authors be sure that other biases are not present in their study, such as being a place that receives many “complicated referrals”, as said by authors? This may also be a reason for the higher mortality rate seen by these authors in comparison to other studies.

3. Figures are blurry and hard to read.

Reviewer #2: Summary:

In this paper, authors analyzed drugs potentially associated with liver injury, including in-hospital mortality and morbidity. In 462 hospitalized patients they found administration of anti-tuberculous drugs to be the most frequent cause of liver injury.

Major comments:

1. Are there any data what is happening out of hospital(s)? In other words how many patients with DILI do not reach hospital at all?

2. Drug-drug interactions are mostly beyond adverse effects of medical treatment in general. Is there any evidence that antituberculotic drugs could have any interactions with other (implicated by authors in Figure 4) or alcohol (despite supposedly not in muslim population). The kind of antituberculotic drugs should be specified including duration of treatment.

3. The in-hospital treatment is of importance. Potential differences between groups under study in in-hospital treatment should be described and/or at least discussed.

Minor comments:

1. Was there any serious bleeding caused by coagulopathy?

2. Was there any information regarding social status?

Conclusion:

Interesting data are presented regarding serious side effects of selected drugs on the liver function in country with supposedly very rare intake of alcohol (in contrast to most of European … countries). Some points need to be addressed more precisely/data added. Please see above.

6. PLOS authors have the option to publish the peer review history of their article (what does this mean?). If published, this will include your full peer review and any attached files.

Reviewer #1: No

Reviewer #2: No

---

## [Author Response · Author response to Decision Letter 0]

20 Mar 2020

Following are our point by point response to academic Editors and reviewers comments. 

Academic Editor:

1. This has been addressed with changes in the revised manuscript files. been included in the naming of the files.

2. The ethics committee waived requirement for informed consent on the grounds of this study being retrospective. Data was not made anonymous to the authors collecting data, but was anonymized for the statistician. This information has been included in the methods section of the manuscript.

 3. These concerns have been addressed with changes in the revised manuscripts (for reference, please see file “Manuscript with Track Changes” page 4 lines 79-82, page 7 lines 145-147 and page 11 lines 231-232)

 4. The list of authors has been amended so that each author is linked to an affiliation and the corresponding author has been identified with an *.

5. Figures 2-4 have been reconstructed to ensure details are clear. 

6. 

The statement "The funders had no role in study design, data collection and analysis, decision to publish, or preparation of the manuscript" has been included in the cover letter.

Reviewer #1: We appreciate the reviewer’s identification of this point excluded from our study. However, the aim of this study is to emphasize DILI of idiosyncratic nature, as intrinsic DILI (such as that caused by acetaminophen in a dose-dependent and predictable manner) has been investigated and reproduced in animal models to elicit a more thorough understanding of liver injury in such cases.

Fontana R. J. (2014). Pathogenesis of idiosyncratic drug-induced liver injury and clinical perspectives. Gastroenterology, 146(4), 914–928. https://doi.org/10.1053/j.gastro.2013.12.032

2. It is very possible that “complicated referrals” and differences in infectious disease epidemiology both contribute to the ATDs becoming the most frequent-causing DILI drug in this study population. We have not come across any studies outlining established correlation between the frequency of DILI causing drugs and disease epidemiology of particular regions. We have amended the language in the manuscript to make note of the above points.

3. Figures have been reconstructed and submitted. 

Reviewer #2: 

1. There is currently no way to quantify the proportion of DILI cases (suspected or definite) that do not reach the hospitals. 

2. Physicians and all healthcare professionals involved in patient care are trained and expected to take a thorough history, regardless of religious or cultural differences. However, the hospital does not have a routine screening questionnaire for alcohol use. Experientially, many Muslims hesitate admitting any alcohol use, which leaves room only for speculation. 

Currently, all patients are given ATDs in the form of Myrin (which contains isoniazid, rifampicin, ethambutol, and pyrazinamide). We have included a table detailing what medications patients were taking besides ATDs, if any (table 2)and a modified figure 4 in the Results section.

3. The patients were managed as per standard of care, essentially supportive treatment was administered. A subset of patients with severe DILI received N- acetyl cysteine. A subset of patients received ventilator support. This information has been added to the Discussion section. 

Minor comments:

1. Was there any serious bleeding caused by coagulopathy?

No

2. Was there any information regarding social status?

No. Aga Khan University Hospital being a tertiary care hospital, caters to patients of all social strata.

---

## [Editor Report · Decision Letter 1]

24 Mar 2020

Drug induced liver Injury is associated with high mortality - a study from a tertiary care hospital in Pakistan

PONE-D-19-34738R1

Dear Dr. Abid,

We are pleased to inform you that your manuscript has been judged scientifically suitable for publication and will be formally accepted for publication once it complies with all outstanding technical requirements.

With kind regards,

Chiara Lazzeri

Academic Editor

PLOS ONE
---

## [Editor Report · Acceptance letter]

26 Mar 2020

PONE-D-19-34738R1 

Drug induced liver Injury is associated with high mortality - a study from a tertiary care hospital in Pakistan 

Dear Dr. Abid:

I am pleased to inform you that your manuscript has been deemed suitable for publication in PLOS ONE. Congratulations! Your manuscript is now with our production department. 

With kind regards,

on behalf of

Dr. Chiara Lazzeri 

Academic Editor

PLOS ONE